artificial intelligence/computer vision/pedology

computer tomography image analysis, deep learning, transfer learning, soil structure analysis, porosity

# Use of deep learning for structural analysis of computer tomography images of soil samples

Ralf Wieland[1], Chinatsu Ukawa[2], Monika Joschko[1], Adrian Krolczyk[1], Guido Fritsch[3], Thomas B. Hildebrandt[3], Olaf Schmidt[4], Juliane Filser[5] and Juan J. Jimenez[6]

[1]Leibniz Centre for Agricultural Landscape Research, Eberswalder Str. 84, 15374 Müncheberg, Germany
[2]Department of Food and Energy Systems Science, Graduate school of Bio-Applications and Systems Engineering, Tokyo University of Agriculture and Technology, Tokyo, Japan
[3]Leibniz Institute for Zoo and Wildlife Research, Reproduction Management, Berlin, Germany
[4]UCD School of Agriculture and Food Science, University College Dublin, Belfield, Dublin, 4, Ireland
[5]University of Bremen, UFT, Department of General and Theoretical Ecology, Bremen, Germany
[6]ARAID, IPE-CSIC, ES, Department of Biodiversity Conversation and Ecosystem Restoration, Jaca, Spain

RW, 0000-0002-2278-610X; CU, 0000-0002-9103-1392; MJ, 0000-0002-4160-1481; AK, 0000-0002-1815-3908; TBH, 0000-0001-8685-4733; OS, 0000-0003-0098-7960; JF, 0000-0003-1535-6168; JJJ, 0000-0003-2398-0796

**Author for correspondence:**
Ralf Wieland
e-mail: rwieland@zalf.de

Soil samples from several European countries were scanned using medical computer tomography (CT) device and are now available as CT images. The analysis of these samples was carried out using deep learning methods. For this purpose, a VGG16 network was trained with the CT images (X). For the annotation (y) a new method for automated annotation, 'surrogate' learning, was introduced. The generated neural networks (NNs) were subjected to a detailed analysis. Among other things, transfer learning was used to check whether the NN can also be trained to other y-values. Visually, the NN was verified using a gradient-based class activation mapping (grad-CAM) algorithm. These analyses showed that the NN was able to generalize, i.e. to capture the spatial structure of the soil sample. Possible applications of the models are discussed.

# 1. Introduction

The analysis of the soil structure by means of CT as an imaging method has opened up a completely new view of soil properties and the impacts of agriculture management on the soil. A lot of processes in the soil (storage of air, water, nutrients, the transport of water and dissolved nutrients, etc.) depend on the soil profile [1]. A crucial step in this work is to extract the relevant soil structures from the CT images [2]. For this purpose, the soil parameters are extracted from CT images by various methods, as discussed in [3]. One step further to a three-dimensional analysis, including a multi-fractal analysis, was introduced in [4]. This underlines that a three-dimensional (3D) analysis is essential for an understanding of the CT images and therefore of the soil structure. The 3D visualization as part of the 3D analysis allows the representation of soil constituents such as roots, earthworm burrows, stones etc. Figures 1 and 2 show screen shots with a 3D image of stones and the same 3D image with stones and wood.

The 3D images have a (Y,Z)-resolution of $0.351 \times 0.351$ mm with a (X)-slice thickness of 0.25 mm. Thus the soil is segmented and understood as a 3D structure via the individual layers. The 3D visualization can be used to quantify the impact of different management practices such as fertilization, plant protection, soil cultivation with or without ploughing etc. In the structural analysis explained below, metrics are generated from the 3D images, which can be used both to describe the soil as well as for machine learning.

Artificial intelligence (AI) methods such as machine learning [5] and especially deep learning [6] promise new ways of analysing CT images. However, it is not known currently to what extent deep learning can be used to analyse soil structures. A promising path was developed in [7] applying a fully convolutional networks to learn pore characteristics.

A key challenge for the use of deep learning is annotation. Annotation means that each image (2D/3D) is individually assigned a value, which is then learned. This is easy to implement, for example, in autonomous driving or in the automated detection of carcinomas in X-ray images [8], since the results can be used by other institutions and thus far exceed the costs of annotation. By contrast, for the scientific analysis of 2D/3D data this effort is often not affordable. This is especially true for the analysis of soil structure, as there is a great scientific interest but financial resources are limited. A possible way out of this conundrum has been suggested in [9]. These authors used a convex envelope to transform the annotation from 3D to 2D space and thus simplify the training of the neural network (NN). In the following, an alternative approach called 'surrogate' learning is presented by calculating the porosity of selected objects like organic material, soil matrix, stones etc. for CT images slice by slice and using this as annotation. Porosity in soil science refers to soil pores, but we have used a wider definition of porosity $P_{slice}$,

$$P_{slice} = \frac{\text{Area(void)}}{\text{Area(void)} + \text{Area(solid)}} \tag{1.1}$$

The porosity is used here as an abstract value consisting of void and solids parts (equation (1.1)).
Two scientific questions arise, which will be investigated step by step:

— Is it possible to use the easy-to-calculate porosity to capture the structure of the soil using a deep learning algorithm?
— Do the models generated by deep learning generalize, or is over-training dominant?

These questions are addressed below, so the paper has a clear focus on AI methods. It should be emphasized again that only deep learning can capture spatial structures. These structures are stored in different resolutions in the layers of the deep learning model.

# 2. Methods

## 2.1. Data

The soil samples (from forest and grassland sites) were collected within the framework of a COST Action ES1406 (KEYSOM) [10] in a number of European countries. For this purpose, intact soil cores were extracted in cylinders with a diameter of 12 cm and a height of 12 cm using a device. The device ensures that the soil sample is undisturbed. Field sampling is illustrated in figures 3 and 4. The

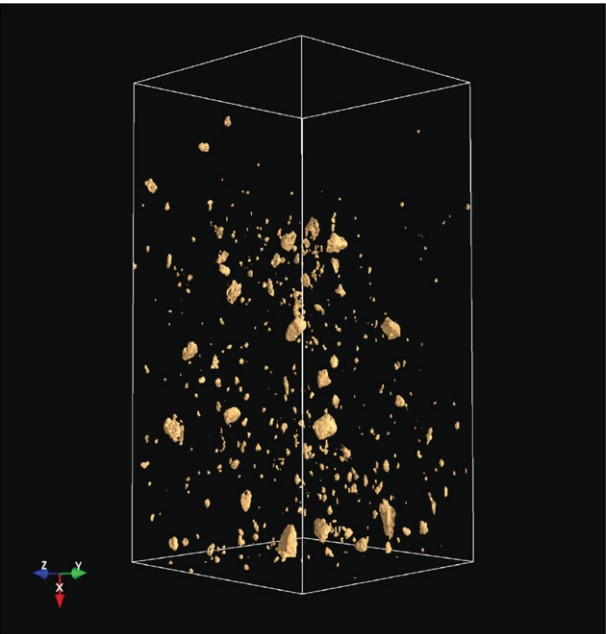

**Figure 1.** Stones in a 3D CT image.

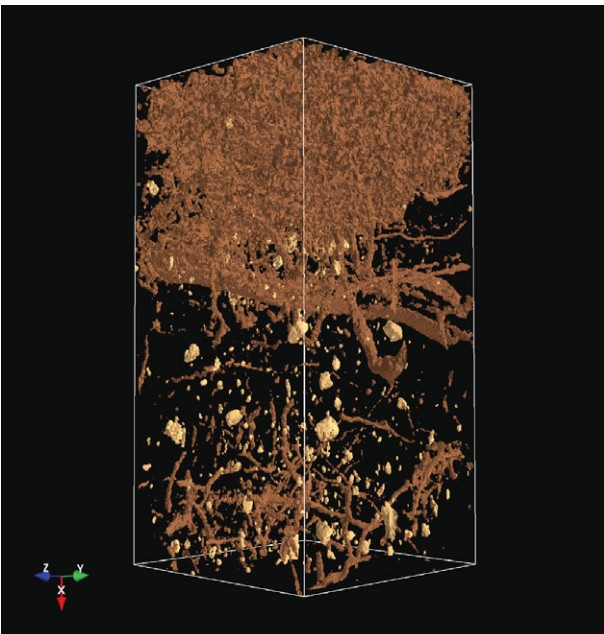

**Figure 2.** Stones and wood in a 3D CT image.

samples were scanned on a 'TOSHIBA Aquilion ONE Genesis' at the Leibniz Institute for Zoo and Wildlife Research (IZW). The dataset is available in DICOM format (Digital Imaging in Medicine).

The image data in DICOM format was created with the Python software 'pydicom'[1] into binary arrays without loss. The values of the arrays are so-called Hounsfield units, which represent a linear transformation of the attenuation of X-rays in tissue. The scale is theoretically open at the top but is limited in medicine to a range of [−1024 HU, 3071 HU]. Stones can also have higher HU, so that the range of [−2048 HU, 4096 HU] was chosen. In the following procedure, a constant of 2048 shifted the numerical range to [0,6144].

---

[1]See https://pydicom.github.io/pydicom/stable/.

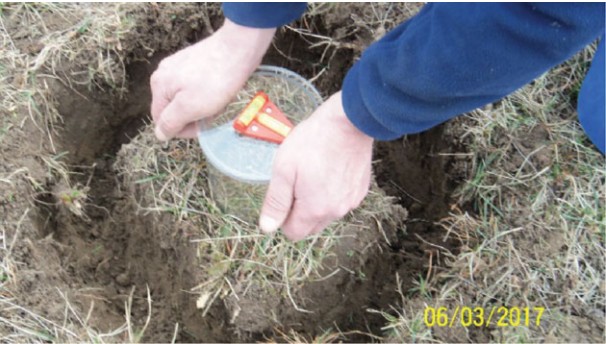

**Figure 3.** Equipment consisting of a pressure plate.

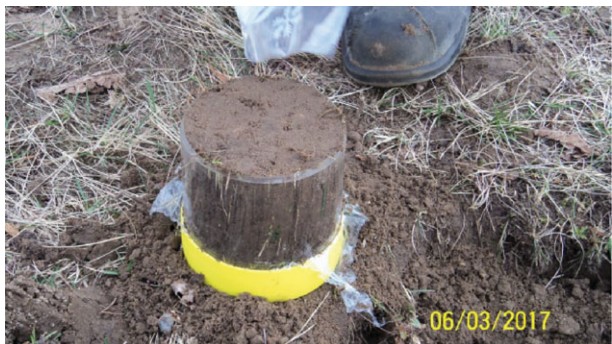

**Figure 4.** Undisturbed soil sample with 12 cm diameter.

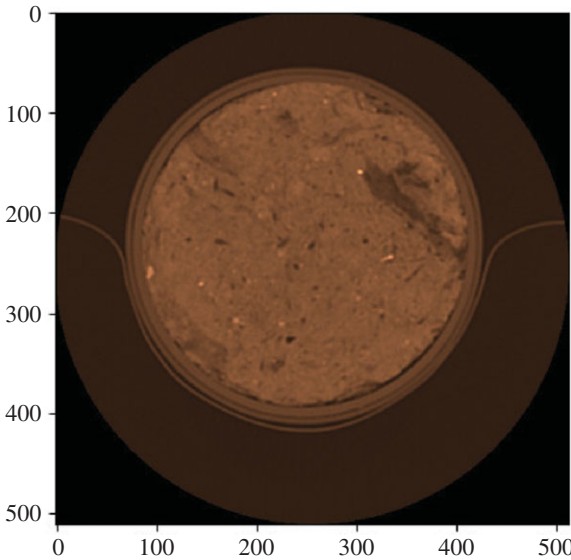

**Figure 5.** Slice with a diameter of 12 cm.

The CT has a yz resolution of 0.351 × 0.351 mm with a slice thickness of 0.25 mm, and 512 rows and 512 columns are generated; the number of slices was 640. To avoid edge effects caused by the cylinder, the volumes were reduced to the inner range of $[X, Y, Z] = [400, 200, 200]$ for further analysis. Figures 5 and 6 show an example of a disc and its square section.

Subsequently, the data will always refer to volumes with the orientation (X,Y,Z).

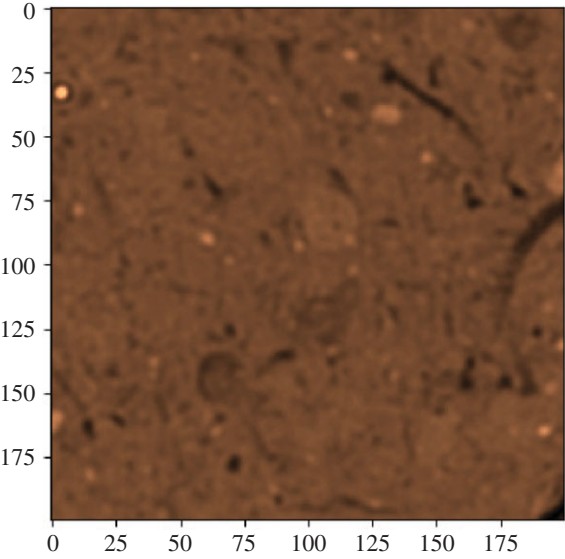

**Figure 6.** Square.

## 2.2. Structure analysis

Important soil processes take place in the pores [11,12]. A comprehensive numerical analysis is presented in [13]. Some of these ideas are taken up in the following and implemented with the Python software 'porespy'.[2] Porespy consists of modules, with the modules 'filters' and 'metrics' being particularly important for pore analysis. In the filters, 2D/3D data are processed so that they can be transformed into values for machine learning using the metrics. The filter 'porespy.filters.trim_small_clusters' removes small pores that are not relevant. For our purposes, a root filter less than 60 voxels was used. The metric 'porespy.metrics.porosity' is used to calculate the porosity of a disc, the function 'porespy.metrics.porosity_profile' calculates the profile over a volume.

Porespy works in such a way that pores are coded with 0 and solid parts with 1. A pore can be filled with air or water, but can also be filled with roots (see the definition of porosity equation (1.1)). It is therefore important to define what is considered solid and what is considered permeable. In order to proceed systematically, the entire data range of [0,6144] has been divided into sub-ranges: [0,1023], [1024,2047], [2048,3071] etc. and titled with (object class1, object class2,…), in short, object1, object2, etc. To illustrate this, the porosity over each area was calculated separately in a pine forest, where the selected object is defined as 'solid'. Figure 7 shows the porosity of a soil sample from a pine forest over the depth according to the six objects.

For comparison, the porosity of a core from sandy grassland soil is shown in figure 8. with a (Z,Y) resolution of 0.351 × 0.351 mm and a slice thickness (X) of 0.25 mm.

With the help of these and other metrics to be determined with porespy, the soil samples can be evaluated and can serve as a basis for a model for machine learning. Returning to the research questions, we will further investigate how deep learning, which uses the porosity of individual objects as a proxy, learns.

## 2.3. Deep learning

A number of proven network structures are available for deep learning, of which VGG16 [14] has proven to be particularly suitable. It is a good compromise between the complexity of the model and thus training time and achievable model quality expressed as root mean squared error (RMSE). The main idea is to train the network using a simple model to calculate function, the porosity in this case, but the network should learn the general structure of the soil sample.

In the structure of VGG16 (figure 9), the classification layer (1 × 1 × 1000) has been replaced by a (1 × 1 × 1) layer, so that porosity can be used as a training target. The idea is that the convolutional

[2]See https://github.com/PMEAL/porespy.

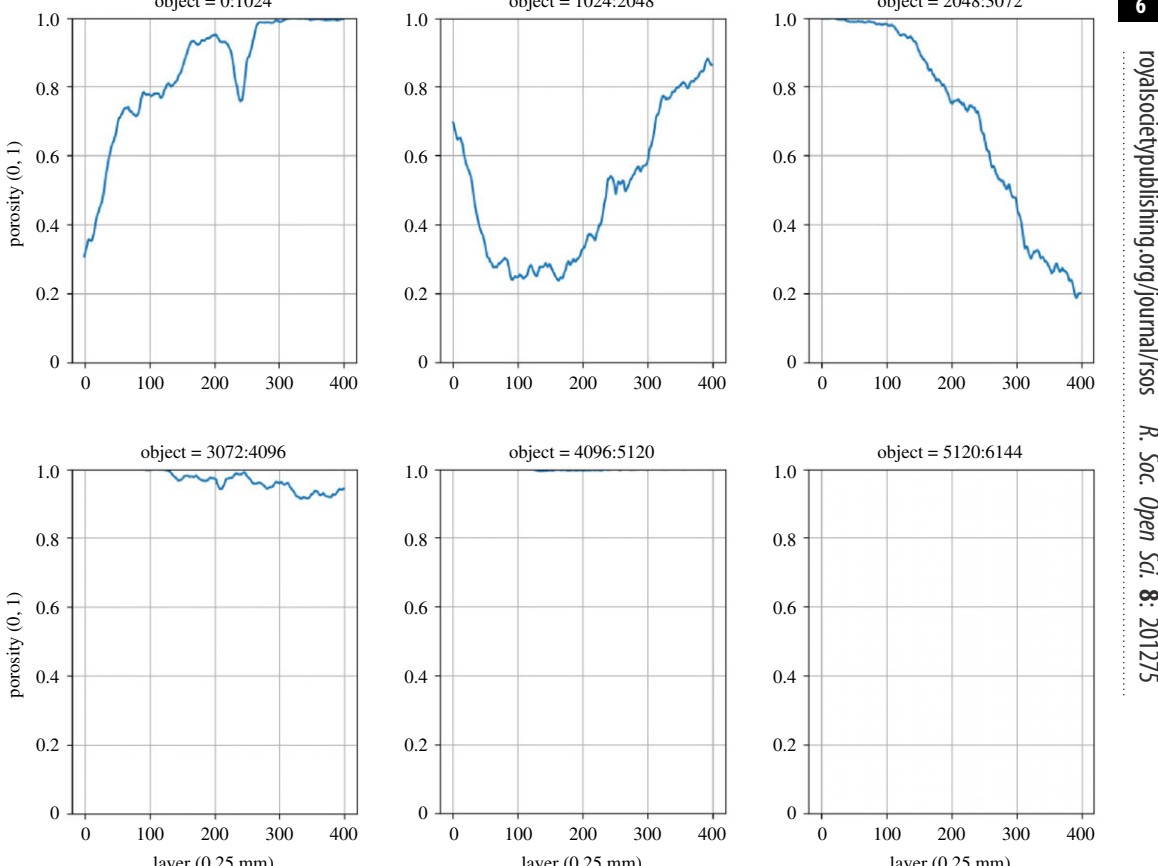

**Figure 7.** Porosity of a pine forest (sample 3243) according to six objects.

neural network (CNN) layers [15] are trained so that the model can be used for other tasks. In the transfer learning [16] used below, only the dense layers (fully connected layer) are trained, but the CNN layers are kept constant. This is often combined with a fine-tuning step [17]. In this context, successful transfer learning would indicate a generalized CNN structure.

Figure 10 shows an input with three layers. This is because the VGG16 requires an RGB image as input. This provides a possibility to structure the information for the training as well.

Figure 11 shows the inputs for the VGG16 and figure 12 the output to be obtained.

NNs are often regarded as black-box models. But especially CNN-based deep learning models can be visualized quite well. On the one hand, the CNNs can be visualized in terms of inputs, so that it is possible to understand what the CNNs see. However, a more interesting method is to observe which part of the picture is used for decision making. The gradient-based class activation mapping (grad-CAM) algorithm is able to do the task [18]. Programs were developed for both approaches, but here we will only present and discuss grad-CAM.

## 3. Results

The training was conducted on a computer with 4 GPUs (TESLA V100). Thirty-two soil samples were trained, each with four different outputs (porosity 1,2,3,5+6). The results were verified by transferring a model trained with the porosity of object3 to a training of the dense layer using the porosity of object2 (transfer learning). The CNN layers are untouched during the transfer learning (see table 1).

Transfer learning did not work satisfactorily in a few examples. This was investigated by applying each trained model to each soil sample. This means that $32 \times 32$ model simulations were performed with $32 \times 32$ RMSE values. The question arose: is there a model that can cope with different datasets than the one it was trained with? Therefore, an iterative procedure was developed to select a model that is compatible with as many soil samples as possible; RMSE < 0.04 was chosen arbitrarily. Then the same procedure was applied to the soil samples that had not been used so far.

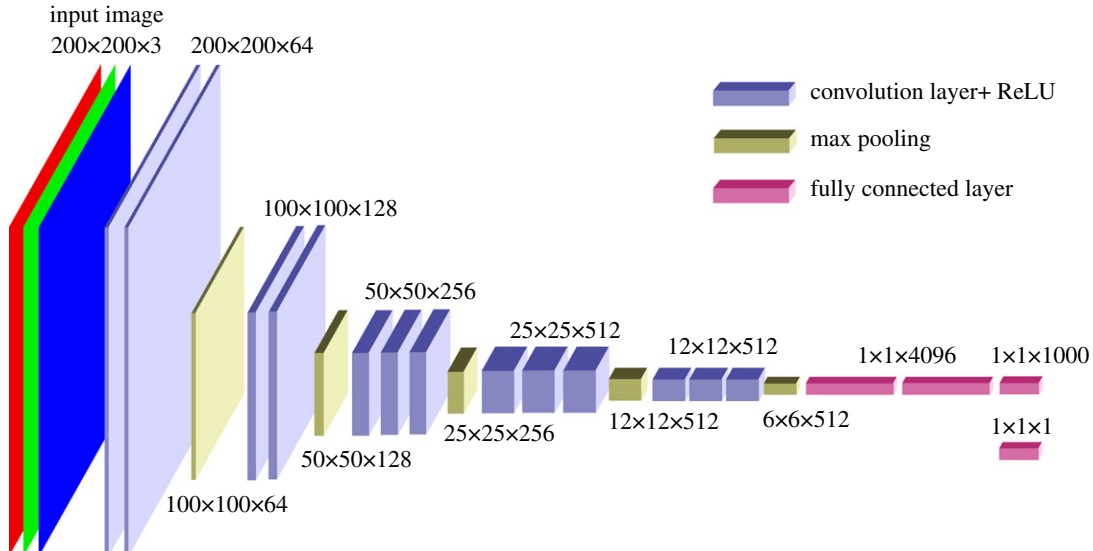

**Figure 8.** Porosity of a grassland (sample 2063) soil after six objects.

**Figure 9.** VGG16 adapted for regression.

Table 2 shows the absolute dominance of the sample 3243 model. A closer look at the soil sample shows that this core is so heterogeneous that it thus contains layers that are similar to those found in other soil samples. By contrast, a grassland site is much less heterogeneous. In figure 12 is shown object1 (organic material) of sample 3243, in figure 13 is shown object1 of the sample 3086.

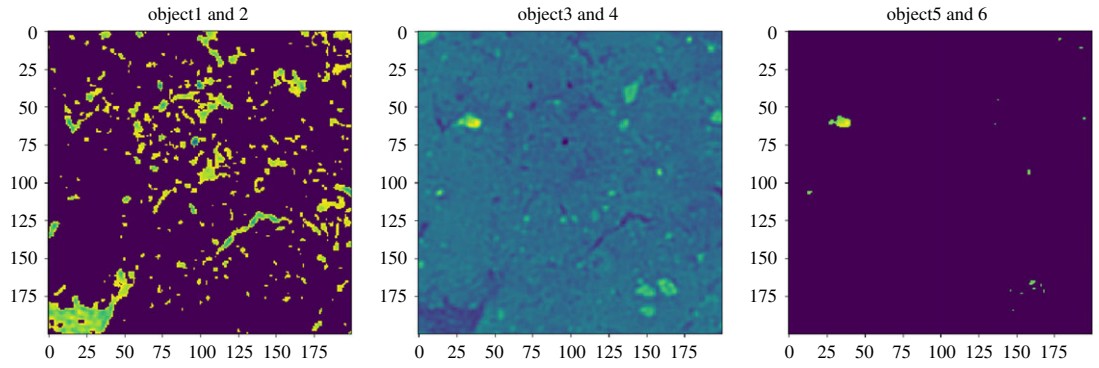

**Figure 10.** Input Images: object1+2, 3+4, 5+6.

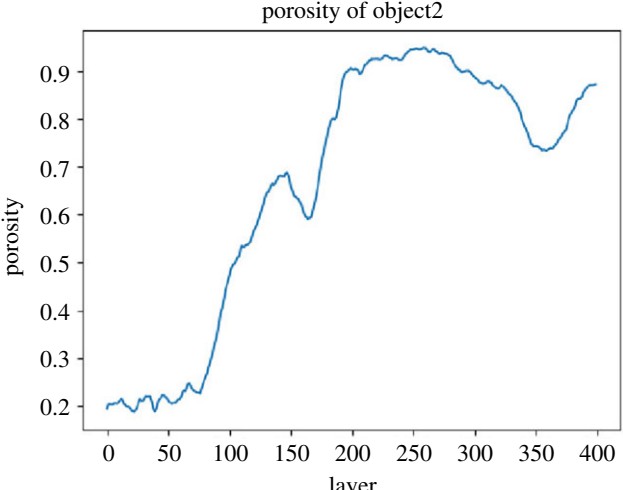

**Figure 11.** Output: Porosity of object2.

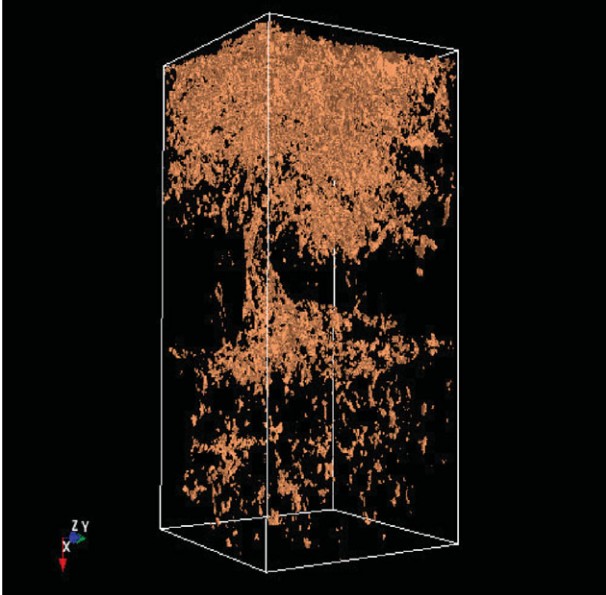

**Figure 12.** Object1 of sample 3243.

In figure 14 is shown object4 (mostly stones) of sample 3243; in figure 15 is shown object4 of sample 3086.

The differences are obvious. However, the neural network also detected that the two soil samples are not compatible.

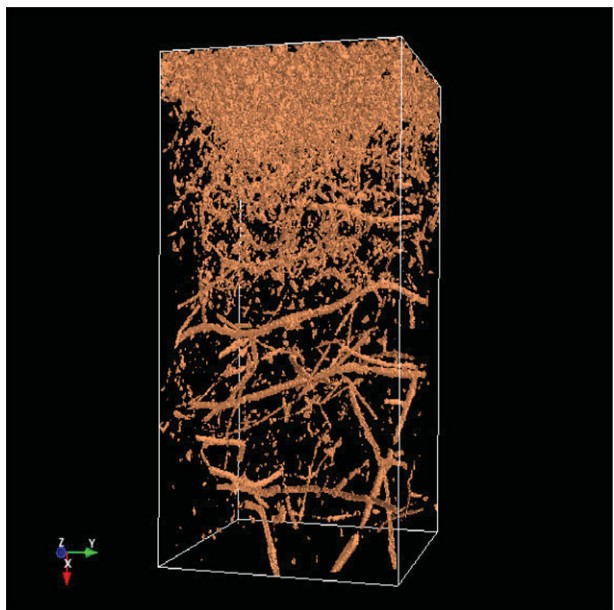

**Figure 13.** Object1 of sample 3086.

**Table 1.** Comparison of computing time for training and transfer learning and the achieved RMSE for the training data (80% randomly selected data) and the test data (20% randomly selected data).

| location | time (s) | train_error | test_error |
|---|---|---|---|
| 3077 | 117.50 | $7.53 \times 10^{-5}$ | $4.40 \times 10^{-5}$ |
| dense layer | 58.77 | $6.17 \times 10^{-5}$ | 0.000118 |
| 3081 | 113.32 | 0.000151 | 0.000293 |
| dense layer | 59.59 | $9.50 \times 10^{-5}$ | 0.000187 |
| 3237 | 111.74 | 0.000463 | 0.000519 |
| dense layer | 62.71 | 0.000748 | 0.00126 |

**Table 2.** Summary RSME (model(data index))<0.04.

| model (sample) | number of similar datasets |
|---|---|
| 3243 | 15 |
| 3064 | 4 |
| 3237 | 4 |
| 3087 | 3 |
| 3078 | 2 |
| 3182 | 1 |
| 3086 | 1 |
| 3242 | 1 |
| 3066 | 1 |

The heat map (figure 16) shows very clearly that the model is focused on pores. The heat maps can be visualized for all CNN layers and all inputs and support the understanding of the CNN's mode of operation.

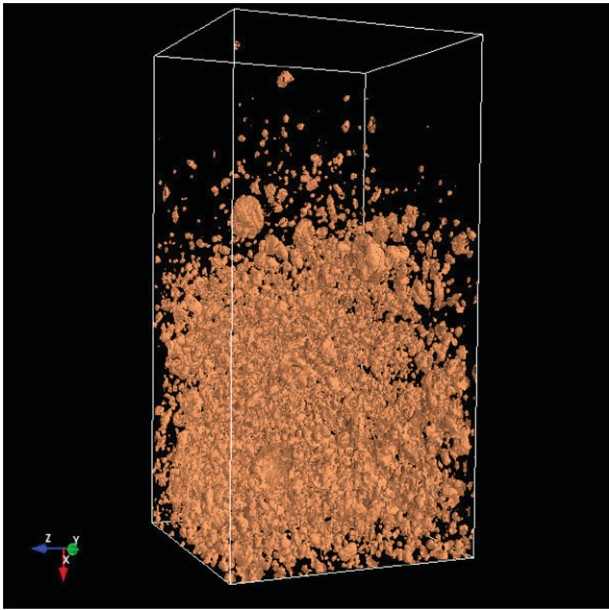

**Figure 14.** Object4 of sample 3243.

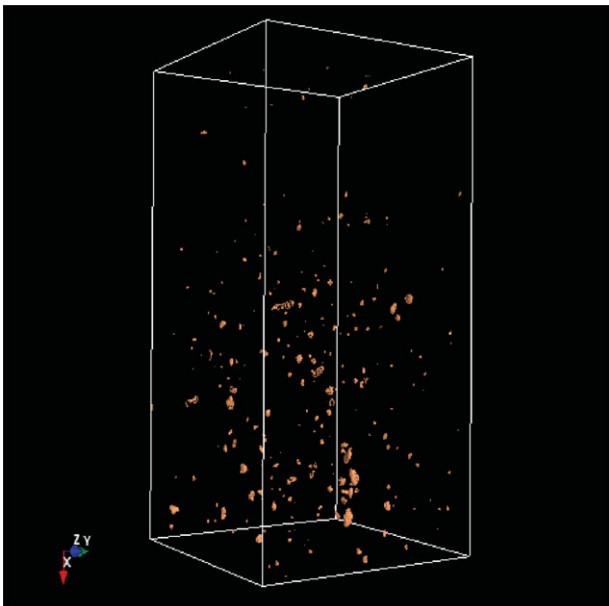

**Figure 15.** Object4 of sample 3086.

# 4. Discussion

A surrogate learning was presented, which extracts the annotation for the training of the NN from an easily obtained function, here porosity. The goal was to create an NN that captures the characteristics of the underlying data structure in its CNN part. This makes it possible to train the NN to other outputs by transfer learning. Using the grad-CAM algorithm it could be visualized that the NN captures the important areas of the soil sample. This reduced a very time-consuming step, the annotation, to a simple learning of a function. Normally, complex deep learning structures as shown in [19,20] are used for automated annotation. Whether the method of surrogate learning introduced here is also useful for other problems is still open.

To reduce the over-training of a deep learning NN, the usual methods such as separation into training set and validation set or early stopping [21] were used. Further measures for regularization are described in

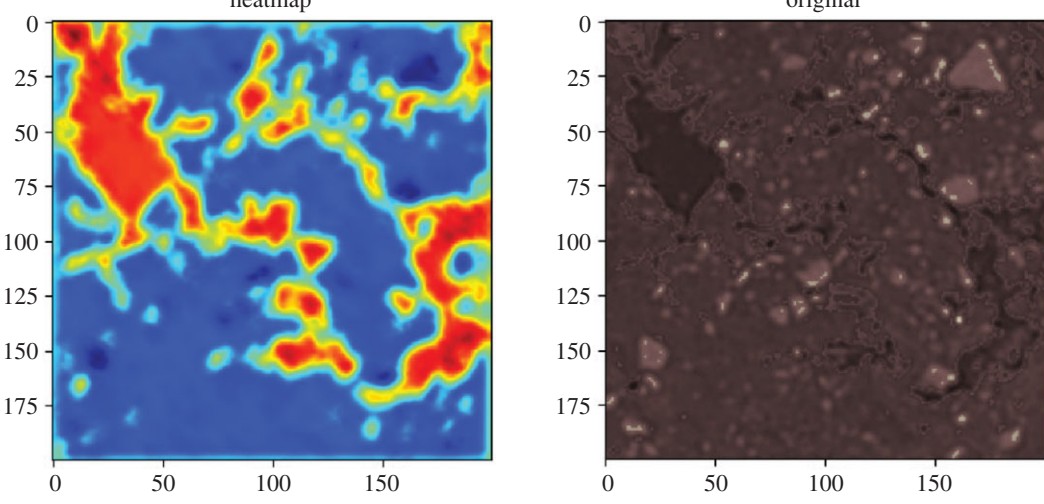

**Figure 16.** Heatmap, original.

[22]. Transfer learning also contributes to the identification of over-training. The fact that, as in our case, the CNNs can be used for other functions, is a further indication of the generalization of the CNNs. Another noteworthy point is that the visualization of the individual CNNs with the grad-CAM algorithm increases the confidence in the generalization of the CNNs. Even though over-training can never be completely excluded, there is some evidence that it does not negatively influence the usage here.

A few remarks about the hardware and software. Without powerful hard and software, training of far more than 100 models would not be feasible. The above-mentioned cluster of 4 GPUs was used under the open-source software 'Tensorflow' from Google including 'Keras' [23]. We are convinced that it is only through the interaction of hardware and open-source software that deep learning can be successfully used outside the commercial environment. The hardware also includes the CT technology used. In the future, in addition to the clinical CT with rather coarse resolution [24], so-called micro-CTs from materials research [25], which allow a much finer resolution, should also be used. The latter is important because soil processes also occur on the fine-scale (approx. 40 µm).

Considering wider applications of the models, the summary shown in table 2 is very interesting. One can see from this summary a possibility to classify the models. The inherent idea here is to record the training data in the future according to locations with 'healthy' soils and 'damaged' soils. Of course, this also depends on site conditions (soil type) and on the use of the sites, i.e. whether they are arable land, grassland or forest, and their specifics. Our focus here is on grassland and arable land sites. For an expert in soil science, it should be easy to make such a classification. This makes it possible to evaluate soils by recording the soil structure.

To develop the idea of classifying soil structure using deep learning, a much larger database is needed than the 32 soil samples collected in the present COST Action. At least, 300 soil samples should be measured for the classification of arable or grassland sites. This is a very rough estimate and should allow a statistically significant classification between healthy and damaged soils. The collection of soil samples with respect to soil types, arable crops, and arable management should not be carried out randomly, but 'target-related'. Starting with important arable crops, such as winter wheat or maize; important management practices, such as fertilization, plant protection, tillage, etc, sampling should be extended gradually to less dominant crops and practices. This has the advantage that the necessary tests for the development of the models can be carried out at the same time as the ongoing collection of data.

It should not remain unmentioned that the CT images of the soil samples can also be used for the further development of plant growth models [26]. As was mentioned in the Introduction, many soil processes take place at the level of the pores. This opens up opportunities to improve and better parametrize plant growth models.

## 5. Conclusion

A newly introduced technique of 'surrogate' learning was applied successfully to CT images of soil samples. This resulted in a reduction of the time-consuming task of annotation for the specific task of structural analysis. Especially interesting was the fact that some models worked well for a set of soil

samples, but other soil samples were fitted with only a few models. This made it possible to classify both soil samples according to their 'exclusivity', which in practical applications should allow classification into 'healthy' and 'damaged' soils.

Ethics. The authors declare that they have no known competing financial interests or personal relationships that could have appeared to influence the work reported in this paper.

Data accessibility. The data are available on request from Zalf. The complete dataset is stored using Dryad Digital Repository: doi:10.5061/dryad.h44j0zpjf [27]. The scripts are located under: https://doi.org/10.5281/zenodo.4540387.

Authors' contributions. R.W. conceptualization, methodology, software, writing—original draft; C.U. software, visualization; M.J. supervision, conceptualization; A.K. data curation, formal analysis; G.F. resources; T.H. resources; O.S. writing—review & editing; J.F. funding acquisition, project administration; J.J.J. funding acquisition, project administration.

Competing interests. We declare we have no competing interests.

Funding. No funding has been received for this article.

Acknowledgements. This article is based upon work from COST Action ES1406 (KEYSOM), supported by COST (European Cooperation in Science and Technology). We thank all the contributors from the COST network for sending us soil samples for this analysis. We thank the Landwirtschaftliche Rentenbank Frankfurt/Main for supporting this study within the DIWELA project (Joschko et al. 2020). Thanks are also due to Holger Schultz and Marcel Paschen from the UGT company Müncheberg for technical support.

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
