## [Peer Review File · Royal Society Open Science]

Review History

RSOS-201275.R0 (Original submission)

Review form: Reviewer 1

Is the manuscript scientifically sound in its present form?

Yes

Are the interpretations and conclusions justified by the results?

Yes

Is the language acceptable?

Yes

Do you have any ethical concerns with this paper?

No

Have you any concerns about statistical analyses in this paper?

No

Recommendation?

Accept with minor revision (please list in comments)

Comments to the Author(s)

The authors proposed a nice work to capture soil structure by deep learning based on computer tomography (CT) images of soil samples from several European countries. The study introduced novel techniques to enhance the ability of artificial intelligence (AI) to detect soil structure metrics, which would be interested for both soil and AI scientists. Some comments are as follows.

Authors could provide some comments on the advantages of deep learning comparing to other machine learning methods. The necessity and significance of deep learning for soil structure analysis should be further highlighted.

At the beginning of the Introduction, some background and importance of soil structure should be mentioned.

The first paragraph of Results seems more like methods, which could be considered to move the Methods. Similarly, the first mentioning of grad-CAM should also be done in Methods.

Some figures miss units on axes, like Figs. 7 and 8. Some similar figures could be combined into a single one like Figs 1 and 2, Figs 12 and 13.

The language should be further checked, for example, in the third paragraph of page 4, the "land" should be removed in "grassland land soil ...".

Review form: Reviewer 2

Is the manuscript scientifically sound in its present form?

Yes

Are the interpretations and conclusions justified by the results?

Yes

Is the language acceptable?

Yes

Do you have any ethical concerns with this paper?

No

Have you any concerns about statistical analyses in this paper?

No

Recommendation?

Accept with minor revision (please list in comments)

Comments to the Author(s)

general remarks:

- very pragmatic approach in the interdisciplinary field between soil science and AI learning methods

- clear formulation of the question
- methods: very brief but well comprehensible; more detailed description for clarification of the research question, however, not at all necessary
- results very well understandable; only 32 samples are critical for a serious discussion of the results. However, the authors themselves cite this point of criticism. But: As a feasibility study, the paper impressively shows the potential of the described approach.

minor changes proposed:

1. p.7line13 "a proxy for (y)" ??? sorry, I do not understand
2. structure analysis: explained in prose only, some formalization would make this step more clear
3. in the figures: add precise scale and description to the axis; give some more words for the figure titles!

Decision letter (RSOS-201275.R0)

Dear Dr Wieland

On behalf of the Editors, we are pleased to inform you that your Manuscript RSOS-201275 "Use of Deep Learning for structural analysis of CT-images of soil samples" has been accepted for publication in Royal Society Open Science subject to minor revision in accordance with the referees' reports. Please find the referees' comments along with any feedback from the Editors below my signature.

Please submit your revised manuscript and required files (see below) no later than 7 days from today's (ie 08-Feb-2021) date. Note: the ScholarOne system will 'lock' if submission of the revision is attempted 7 or more days after the deadline. If you do not think you will be able to meet this deadline please contact the editorial office immediately.

Best regards,
Lianne Parkhouse
Editorial Coordinator

on behalf of the Associate Editor and Professor Peter Haynes (Subject Editor)
openscience@royalsociety.org

Associate Editor Comments to Author:

Thank you for your patience with the unusually long review period: no doubt the COVID pandemic has been a contributing factor, but we regret the difficulty the editors have had in finding referees - a larger-than-normal number of invitations needed to be sent to secure the two commentaries we have received. We hope these comments are useful in revising your paper, and we'll look forward to the revision in due course.

Reviewer comments to Author:

Reviewer: 1

Comments to the Author(s)

The authors proposed a nice work to capture soil structure by deep learning based on computer tomography (CT) images of soil samples from several European countries. The study introduced novel techniques to enhance the ability of artificial intelligence (AI) to detect soil structure metrics, which would be interested for both soil and AI scientists. Some comments are as follows.

Authors could provide some comments on the advantages of deep learning comparing to other machine learning methods. The necessity and significance of deep learning for soil structure analysis should be further highlighted.

At the beginning of the Introduction, some background and importance of soil structure should be mentioned.

The first paragraph of Results seems more like methods, which could be considered to move the Methods. Similarly, the first mentioning of grad-CAM should also be done in Methods.

Some figures miss units on axes, like Figs. 7 and 8. Some similar figures could be combined into a single one like Figs 1 and 2, Figs 12 and 13.

The language should be further checked, for example, in the third paragraph of page 4, the "land" should be removed in "grassland land soil ...".

Reviewer: 2

Comments to the Author(s)

general remarks:

- very pragmatic approach in the interdisciplinary field between soil science and AI learning methods
- clear formulation of the question
- methods: very brief but well comprehensible; more detailed description for clarification of the research question, however, not at all necessary
- results very well understandable; only 32 samples are critical for a serious discussion of the results. However, the authors themselves cite this point of criticism. But: As a feasibility study, the paper impressively shows the potential of the described approach.

minor changes proposed:

1. p.7line13 "a proxy for (y)" ??? sorry, I do not understand
2. structure analysis: explained in prose only, some formalization would make this step more clear
3. in the figures: add precise scale and description to the axis; give some more words for the figure titles!

===PREPARING YOUR MANUSCRIPT===

===PREPARING YOUR REVISION IN SCHOLARONE===

<https://royalsociety.org/journals/authors/author-guidelines/#supplementary-material> to include a suitable title and informative caption. An example of appropriate titling and captioning may be found at https://figshare.com/articles/Table_S2_from_Is_there_a_trade-off_between_peak_performance_and_performance_breadth_across_temperatures_for_aerobic_sc_ope_in_teleost_fishes_/3843624.

Author's Response to Decision Letter for (RSOS-201275.R0)

See Appendices A - C.

Decision letter (RSOS-201275.R1)

Dear Dr Wieland,

It is a pleasure to accept your manuscript entitled "Use of Deep Learning for structural analysis of CT-images of soil samples" in its current form for publication in Royal Society Open Science. The comments of the reviewer(s) who reviewed your manuscript are included at the foot of this letter.

on behalf of Peter Haynes (Subject Editor)
openscience@royalsociety.org

Appendix A

Dear Liane, dear Andrew,

thank you both for your great support during the submission process. Only with your help was I able to complete the work. I have revised the work according to the suggestions of the reviewers. They have also done a great job. Please also address my thanks to Professor Peter Haynes for his supervision of the revision process.

Dear Reviewer: 1,

many thanks for your review and your helpful comments. They helped me a lot to make the paper better.

Comments to the Author(s)

The authors proposed a nice work to capture soil structure by deep learning based on computer tomography (CT) images of soil samples from several European countries. The study introduced novel techniques to enhance the ability of artificial intelligence (AI) to detect soil structure metrics, which would be interested for both soil and AI scientists. Some comments are as follows.

Authors could provide some comments on the advantages of deep learning comparing to other machine learning methods. The necessity and significance of deep learning for soil structure analysis should be further highlighted.

- Deep learning is the only way to find the structures in a soil sample that determine its physical properties, in this case the flow and storage of water. I have added a comment in the paper.

At the beginning of the Introduction, some background and importance of soil structure should be mentioned.

- I have added a comment and a citation.

The first paragraph of Results seems more like methods, which could be considered to move the Methods. Similarly, the first mentioning of grad-CAM should also be done in Methods.

- I have moved the grad-CAM into the method section.

Some figures miss units on axes, like Figs. 7 and 8. Some similar figures could be combined into a single one like Figs 1 and 2, Figs 12 and 13.

- I have added the axis.

The language should be further checked, for example, in the third paragraph of page 4, the "land" should be removed in "grassland land soil ...".

- I changed it.

Many thanks again, Ralf Wieland.

Dear Reviewer: 2,

many thanks for your review and your helpful comments. They helped me a lot to make the paper better.

Comments to the Author(s)

general remarks:

- very pragmatic approach in the interdisciplinary field between soil science and AI learning methods
- clear formulation of the question
- methods: very brief but well comprehensible; more detailed description for clarification of the research question, however, not at all necessary
- results very well understandable; only 32 samples are critical for a serious discussion of the results. However, the authors themselves cite this point of criticism. But: As a feasibility study, the paper impressively shows the potential of the described approach.

minor changes proposed:

1. p.7line13 "a proxy for (y)" ??? sorry, I do not understand

- I have changed this

2. structure analysis: explained in prose only, some formalization would make this step more clear

- I have added a ref to the definition of porosity in the introduction.

3. in the figures: add precise scale and description to the axis; give some more words for the figure titles!

- I have added the axis

Many thanks again, Ralf Wieland.

Appendix B

Dear Reviewer: 1,

many thanks for your review and your helpful comments. They helped me a lot to make the paper better.

Comments to the Author(s)

The authors proposed a nice work to capture soil structure by deep learning based on computer tomography (CT) images of soil samples from several European countries. The study introduced novel techniques to enhance the ability of artificial intelligence (AI) to detect soil structure metrics, which would be interested for both soil and AI scientists. Some comments are as follows.

Authors could provide some comments on the advantages of deep learning comparing to other machine learning methods. The necessity and significance of deep learning for soil structure analysis should be further highlighted.

- Deep learning is the only way to find the structures in a soil sample that determine its physical properties, in this case the flow and storage of water. I have added a comment in the paper.

At the beginning of the Introduction, some background and importance of soil structure should be mentioned.

- I have added a comment and a new citation.

The first paragraph of Results seems more like methods, which could be considered to move the Methods. Similarly, the first mentioning of grad-CAM should also be done in Methods.

- I have moved the grad-CAM into the method section.

Some figures miss units on axes, like Figs. 7 and 8. Some similar figures could be combined into a single one like Figs 1 and 2, Figs 12 and 13.

- I have added the axis.

The language should be further checked, for example, in the third paragraph of page 4, the "land" should be removed in "grassland land soil ...".

- I changed it.

Many thanks again, Ralf Wieland.

Appendix C

Dear Reviewer: 2,

many thanks for your review and your helpful comments. They helped me a lot to make the paper better.

Comments to the Author(s)

general remarks:

- very pragmatic approach in the interdisciplinary field between soil science and AI learning methods
- clear formulation of the question
- methods: very brief but well comprehensible; more detailed description for clarification of the research question, however, not at all necessary
- results very well understandable; only 32 samples are critical for a serious discussion of the results. However, the authors themselves cite this point of criticism. But: As a feasibility study, the paper impressively shows the potential of the described approach.

minor changes proposed:

1. p.7line13 "a proxy for (y)" ??? sorry, I do not understand

- I have changed this

2. structure analysis: explained in prose only, some formalization would make this step more clear

- I have added a ref to the definition of porosity in the introduction.

3. in the figures: add precise scale and description to the axis; give some more words for the figure titles!

- I have added the axis

Many thanks again, Ralf Wieland.